# A Comparative Analysis of Photon versus Proton Beam Therapy in Neoadjuvant Concurrent Chemoradiotherapy for Intrathoracic Squamous Cell Carcinoma of the Esophagus at a Single Institute

**DOI:** 10.3390/cancers14082033

**Published:** 2022-04-18

**Authors:** Jin-Ho Choi, Jong Mog Lee, Moon Soo Kim, Youngjoo Lee, Yang-Gun Suh, Sung Uk Lee, Doo Yeul Lee, Eun Sang Oh, Tae hyun Kim, Sung Ho Moon

**Affiliations:** 1Department of Thoracic Surgery, Research Institute and Hospital, National Cancer Center, Goyang 10408, Korea; 13016@ncc.re.kr (J.-H.C.); jongmok@ncc.re.kr (J.M.L.); vsd10@ncc.re.kr (M.S.K.); 2Department of Internal Medicine, Research Institute and Hospital, National Cancer Center, Goyang 10408, Korea; yjlee@ncc.re.kr; 3Proton Therapy Center, Research Institute and Hospital, National Cancer Center, Goyang 10408, Korea; suhmd@ncc.re.kr (Y.-G.S.); sulee126@ncc.re.kr (S.U.L.); ldy8816@ncc.re.kr (D.Y.L.); euns0530@ncc.re.kr (E.S.O.); k2oncon@ncc.re.kr (T.h.K.)

**Keywords:** chemoradiation, neoadjuvant, squamous cell carcinoma, esophageal cancer

## Abstract

**Simple Summary:**

Radiotherapy for an esophageal malignancy results in undesired radiation exposure to nearby critical organs such as the heart, lungs, and spinal cord, leading to unfavorable clinical outcomes. Although only utilized at a few centers, proton beam radiotherapy spares adjacent organs from radiation exposure more effectively than conventional photon radiotherapy. We confirmed that, in terms of sparing adjacent organs from radiation, proton beam radiotherapy, as a modality of neoadjuvant chemoradiotherapy, was significantly superior to conventional photon radiotherapy for locally advanced esophageal squamous cell carcinoma with an even distribution of intrathoracic locations, and may lead to improved clinical outcomes.

**Abstract:**

Background: Proton beam therapy (PBT), as a neoadjuvant chemoradiotherapy (nCRT) modality, is expected to result in better outcomes than photon-based radiotherapy (RT) for esophageal cancer, particularly adenocarcinoma. This study reports the results of nCRT for locally advanced esophageal squamous cell carcinoma (ESCC) using both modalities. Methods: We retrospectively reviewed the records of patients who underwent nCRT for ESCC between 2001 and 2020. A median of 41.4 Gy or cobalt gray equivalents of radiation was delivered using either photons or protons, with concurrent chemotherapy. Dosimetric and clinical parameters were compared between the two groups. Results: Of the 31 patients, the lungs and heart of the proton group (*n* = 15) were exposed to significantly less radiation compared to the photon group (*n* = 16). No significant differences in short-term postoperative outcomes or lymphocyte count were observed between the groups, and there were no significant differences between the photon and proton groups in 2-year overall survival (67.8% vs. 68.6%, *p* = 0.867) or 2-year disease-free survival (33.3% vs. 34.5%, *p* = 0.749), with a median follow-up of 17 months. Conclusions: PBT provided a significant dosimetric benefit over photon-based RT during nCRT for ESCC; however, it did not improve clinical outcomes.

## 1. Introduction

Esophageal cancer (EC) ranks seventh in incidence and sixth in mortality according to recent global reports [1]. The prognosis of EC is poor, with an estimated 5-year survival rate of only approximately 20% because less than 20% of patients are diagnosed at the localized stage [2]. Neoadjuvant chemoradiotherapy (nCRT) followed by surgery improves overall survival (OS) compared to surgery alone, and is widely accepted as the standard treatment for localized operable EC [3]. The incidence of esophageal adenocarcinomas (EACs) is increasing in Western countries, but esophageal squamous cell carcinoma (ESCC) remains the most prevalent type worldwide, accounting for an estimated 84% of all cases [4]. The survival benefit of nCRT is greater for ESCC than EAC. 

In the two-dimensional RT era, it has proven challenging to prepare optimal radiotherapy (RT) plans for patients with EC because the esophagus is surrounded by organs at risk (OAR) including the lungs, heart, and spinal cord. However, recent advances in RT technology have made it possible to deliver an adequate radiation dose with improved conformity to the tumor while also sparing adjacent normal structures. Particle therapies including proton beam therapy (PBT) [5] and carbon ion radiotherapy [6] are next-generation RT technologies designed to spare normal tissues to a greater degree compared with three-dimensional conformal radiotherapy (3D-CRT) and intensity-modulated radiotherapy (IMRT). Available at a limited number of hospitals, PBT for nCRT for esophageal cancers minimizes the radiation dose to OARs compared to photon-based RTs such as 3D-CRT and IMRT [7], thereby reducing the risk of perioperative cardiopulmonary complications [8,9] and severe lymphopenia [10]. Preserving circulating lymphocytes, which are critical in the anti-tumor response, may improve the survival rate for EC and other types of malignancies. nCRT with PBT is also expected to improve the survival rate of EC patients [11,12]. 

In this study, we compared photon-based RT and PBT as nCRT modalities for localized operable intrathoracic ESCC in terms of treatment efficacy and postoperative complications at a single institute.

## 2. Materials and Methods

We retrospectively reviewed the medical records of patients in an RT database who underwent pre-planned nCRT followed by curative resection for ESCC between January 2001 and May 2020. A multidisciplinary tumor board at our institution provided the following indications for nCRT of EC: patients able to tolerate surgery and nCRT; patients whose disease was considered resectable by a board-certified thoracic surgeon; and no distant metastasis. Patients who underwent salvage surgery after definitive RT or CRT were excluded. Data on patient demographics as well as the nCRT, operation, chemotherapy, postoperative outcome, and follow-up status were collected. 

### 2.1. Radiotherapy

All patients were in an arm-up supine position and immobilized by a vacuum cushion. For PBT, an in-house made round-type couch with carbon material was applied. In the previous one third of patients with PBT, 3D-CT based passive scattering PBT planning with an adequate PTV margin considering the patient’s respiration was used. Later, patients underwent 3D-CT or 4D-CT based treatment simulation after checking the patient’s respiration signal. When 4D-CT could not be acquired because the patient’s breathing pattern was irregular or the amplitude was too low, abdominal compression with a bellows type belt was applied without respiration-gating 4D-CT simulation. RT was delivered using megavoltage photon beams or proton beams. In most patients, 1.8 Gy of photon-based RT or the cobalt gray equivalent (CGE) of PBT was delivered once daily, five times per week, up to 41.4 Gy. Gross tumor volume (GTV) was defined as the primary gross tumor volume (GTVp) and metastatic nodal gross tumor volume (GTVn), as revealed by chest computed tomography (CT), esophagoscopy, and 18flurodeoxyglucose-positron emission tomography/computed tomography (18FDG-PET/CT). The clinical target volume (CTV) extended 3.5–4 cm craniocaudally, 0.5–1 cm circumferentially from GTVp, and 0.7–1 cm circumferentially from the GTVn. The planning target volume was defined as the CTV plus 0.5–1 cm. Passive-scattering, uniform scanning, or pencil-beam scanning techniques were used to deliver the PBT. The 3D-CRT and PBT plans were produced using the Eclipse planning system (Varian Medical System, Palo Alto, CA, USA). During the 3D-CRT and PBT planning process, anterior–posterior parallel-opposed fields were usually selected, but anterior–posterior oblique fields or posterior oblique fields were chosen at times to enhance conformality of the plan. The IMRT plan was prepared using the Eclipse or TomoTherapy planning system (Accuray, Sunnyvale, CA, USA). Three to five fields were used with a dynamic multileaf collimator or helical tomotherapy unit. During PBT, patient setup was verified by either 2D X-ray positioning system with the DIPS (IBA, Walloon Brabant, Belgium) or weekly cone-beam CT with the AdaPT insight (IBA) without fiducial marker insertion. When patient showed sudden symptomatic change of either aggravation or improvement during treatment, additional CT simulation for adaptive planning was generally concerned.

### 2.2. Chemotherapy

Two chemotherapy regimens were used for nCRT according to the physician’s preference. One regimen included 4–6 weekly cycles of intravenous carboplatin (area under the curve of 2 mg/mL/min) and paclitaxel (50 mg/m^2^). The other regimen comprised two cycles of intravenous cisplatin (60 mg/m^2^) on day 1 and oral capecitabine (1000 mg/m^2^ twice daily) from days 1 to 14 (at 3-week intervals).

### 2.3. Surgery 

The patients underwent surgery after the reevaluation of chest CT and 18FDG-PET/CT scans 4–6 weeks after completing nCRT, but those with systemic progression after nCRT were excluded from surgery. Esophagectomy was performed either by a minimally invasive approach or thoracotomy on the right side, with the graft prepared by upper median laparotomy. The first choice graft for esophageal reconstruction was the stomach, with intrathoracic or cervical anastomosis performed depending on the intrathoracic location of the esophageal cancer. The colon was preferred to reconstruct the esophagus, with anastomosis of the cervical esophagus performed in patients for whom stomach graft was not available. The thoracic and abdominal regional lymph nodes were dissected routinely, and the cervical lymph nodes were dissected when the findings of the initial evaluation were suspicious.

### 2.4. Follow-Up and Statistics

The patients were followed-up at 3–6-month intervals by chest CT and esophagoscopy, at least twice annually. The clinical endpoints of OS and disease-free survival (DFS) were defined as the interval between the date of initiation of nCRT and the date of death from any cause, and the date of proven disease progression or death, respectively. Recurrence was defined based on radiological or endoscopic findings, or through pathological confirmation. The clinical characteristics and outcomes of the photon-based (3D-CRT and IMRT) nCRT group (Group_photon_) and proton-based nCRT group (Group_proton_) were compared. Patients ineligible for curative resection after nCRT were included in the analysis of nCRT outcomes, but excluded from the analysis of postoperative outcomes.

The statistical analysis was performed using IBM SPSS Statistics (version 28.0.1; IBM Corp., Armonk, NY, USA) and Excel software (version 2202; Microsoft Corp., Redmond, WA, USA). The chi-square or Fisher’s exact test was used to compare categorical variables between the two groups, while the independent Student’s *t*-test or Mann–Whitney test was used for continuous variables. Pearson’s correlation analysis was used to analyze the relationship between the lymphocyte count nadir and dosimetric results. The change trend in lymphocyte count during nCRT was analyzed by repeated-measures analysis of variance. A Kaplan–Meier survival curve was used to estimate OS and DFS, and the log-rank test was used to compare the survival data between the two groups.

## 3. Results

A total of 31 patients with ESCC were scheduled for nCRT followed by surgery. Sixteen and fifteen patients were included in the Group_photon_ and Group_proton_, respectively. The two groups did not show significant differences except for the Eastern Cooperative Oncology Group performance status (ECOG PS), where there were significantly more ECOG PS 1 cases in the Group_photon_. The most common location for the primary tumor was the middle thorax (42%), followed by the upper (29%) and lower thoraxes (29%). The patient characteristics are listed in Table 1.

Table 2 summarizes the dose–volume histogram (DVH) parameters of the lungs, heart, and lymphopenia. The median total radiation dose and PTV (cc) did not differ between the two groups. However, the lungs and heart of patients in the Group_proton_ were exposed to significantly less radiation. Patients in the Group_proton_ showed significantly lower DVH parameters than the Group_photon_ including the volume of the lungs (or heart) exposed to at least X Gy (VXorgan) (%) and the mean lung (or heart) dose (Gy). The lymphocyte count decreased continuously during nCRT, but recovered by the time of the operation with no significant difference seen between the groups (Figure 1). The mean ± SD lymphocyte count nadirs of the Group_photon_ and Group_proton_ were 396.49 ± 156.48 and 388.02 ± 239.22, respectively (*p* = 0.907), and no significant difference was found between the two groups in the incidence of grade 4 lymphopenia (12.5% vs. 20.0%, *p* = 0.654; Table 2). The nadir of the absolute lymphocyte count (ALC) during nCRT was negatively correlated with the mean heart dose (r = −0.383, *p* = 0.037) and V10 heart (r = −0.400, *p* = 0.028), but no significant correlations were detected with the DVH parameters of the lung (Figure 2). 

Three patients did not undergo curative resection after nCRT due to a lack of resectability (in two patients; one in each group) and progression of ESCC during nCRT (one patient in the Group_photon_). Thus, 14 patients in each group were included in the analysis of operative outcomes. No significant group differences were observed in the frequency of the minimally invasive approach, intrathoracic anastomosis, or use of a stomach graft to reconstruct the esophagus. Although the rate of three-field lymph node dissection was not different between the Group_photon_ and Group_proton_ (42.9% vs. 42.9%, *p* = 1.000), a significantly greater median (range) number of lymph nodes was resected in Group_proton_ than Group_photon_ [0 (0–44) vs. 34 (0–66), *p* = 0.001]. No significant group differences in the distribution of pathologic T, N, or M categories, or the rate of pathologically complete response (ypT0N0M0) to nCRT were observed (32%). Among all patients, one in the Group_proton_ died on postoperative day 22 due to colon graft necrosis, with no significant difference seen in the postoperative in-hospital mortality between the Group_photon_ and Group_proton_ (0% vs. 7.1%, *p* = 1.000). Postoperative anastomosis or graft failure and respiratory complications were observed in nine (32%) and 12 (43%) patients, respectively, and no significant difference was detected between the two groups in terms of short-term postoperative complications (Table 3).

With a median 17-month follow-up [95% confidence interval (CI): 2.9–31.1], no significant differences were seen between Group_photon_ and Group_proton_ in 2-year OS (67.8% vs. 68.6%, *p* = 0.867) or 2-year DFS (33.3% vs. 34.5%, *p* = 0.749) (Figure 3). After curative resection, recurrence was seen in eight patients in the Group_photon_ and six in the Group_proton_, with no significant group differences observed in the pattern of recurrence [locoregional recurrence only/locoregional and distant/distant recurrence only; 3 (37.5%)/4 (50.0%)/1 (12.5%) vs. 1 (16.7%)/3 (50%)/2 (33.3%), *p* = 0.762].

## 4. Discussion

This study compared photon-based RT and PBT as nCRT modalities to treat operable intrathoracic ESCC in terms of treatment efficacy and postoperative complications. PBT has been focused on as an nCRT modality for EC because it is advantageous for reducing cardiopulmonary and hematological toxicity compared to photon-based RT, and thus can improve the survival outcomes of patients with EC [10,13,14,15]. In a retrospective study, Wang et al. compared photon-based RT (3D-CRT and IMRT) and PBT in patients undergoing nCRT for EC and found that PBT significantly reduced postoperative pulmonary and gastrointestinal complications compared to 3D-CRT [15]. The same investigators reported that patients with grade 4 lymphopenia showed poorer progression-free survival (PFS) in a series of retrospective studies [10,16]. They reported that PBT was significantly associated with a lower likelihood of RT-induced grade 4 lymphopenia compared to IMRT, suggesting that PBT has the potential to improve the survival outcomes of patients undergoing nCRT for EC by sparing circulating lymphocytes [10,16]. Furthermore, a high ALC nadir of more than 0.35 × 10^3^/µL was significantly correlated with a higher rate of pCR after nCRT for EC, while PBT instead of IMRT was one of the predictors of a high ALC nadir in another study [13]. In a subsequent phase IIB clinical trial comparing PBT and IMRT in nCRT for EC, PBT was significantly superior to IMRT in terms of the primary endpoint of total toxicity burden, which is an integrated index of postoperative complications and various nCRT-associated toxicities, but did not demonstrate superiority in terms of PFS [14]. A phase III randomized clinical trial comparing PBT and IMRT for operable EC (NRG GI-006) is currently in progress. 

Although it is difficult to directly compare the treatment outcomes of our study with those of other studies due to differences in patient characteristics, treatment methods, and outcome measures, the survival outcomes of our study were comparable to the 3-year OS rate of nCRT for EC patients in phase III clinical trials, which was about 60% [4,17]. The incidence rates of pulmonary complications, cardiac complications, anastomotic leakage, and in-hospital mortality were 46%, 21%, 22%, and 4%, respectively, for the nCRT group (*n* = 171) in the CROSS study [18], and 15.0% (pulmonary infection, atelectasis, respiratory failure, acute respiratory distress syndrome), 14.6% (arrhythmia, heart failure, acute coronary syndrome), 8.6%, and 0.5%, respectively, for the nCRT group (*n* = 185) in the NEOCRTEC study [19]. Our study showed similar postoperative adverse outcomes such as pulmonary and cardiac complications, anastomotic leakage, and operative mortality, to those of large-scale clinical trials [4,17,19]. 

The lungs and heart in our Group_proton_ were exposed to significantly less radiation compared to the Group_photon_, similar to the findings of most studies that compared DVH parameters between EC patients undergoing photon-based RT or PBT [5,7,15,20,21,22]. However, we failed to demonstrate any significant differences between the two RT techniques in survival outcomes, the incidence of postoperative complications, or the severity of RT-induced lymphopenia. 

It is necessary to consider the following when interpreting the discrepancies between our results and those of the previous series [10,13,14,15,16]. It should be emphasized that the number of patients included in our study was small. Second, our study was different from the previous series in that all patients had ESCC rather than both ESCC and EAC [10,13,14,15,16]. In the previous study, more than 90% of the patients had EAC and lower thoracic EC, whereas in our study, only about 30% of the EC cases involved the lower thorax, so it is possible that the results of both study cohorts differed in terms of postoperative complication rates. PBT would have the advantage of reducing postoperative complications to a greater degree compared to IMRT in patients with lower thoracic EC, as confirmed in previous studies [15,21]. This is because PBT reduces the radiation doses to the cardiac chambers and coronary arteries more so than IMRT. However, further studies are needed on the advantages of PBT over photon-based RT in patients with upper to middle thoracic EC in terms of postoperative complications after nCRT. The ALC nadir was higher in our study, regardless of group, than in other studies, thereby demonstrating the benefit of PBT over photon-based RT in terms of reducing the ALC nadir (as well as the incidence of grade 4 lymphopenia). The proportion of patients who experienced grade 4 lymphopenia in our study was smaller than in previous studies [10,13,16,23]. Therefore, the advantage of sparing lymphocytes in PBT to photon-based RT in patients with upper to middle thoracic EC needs further investigation. Additionally, the larger total radiation dose (50.4 Gy) in the previous series compared to the median total dose (41.4 Gy) in this study was also notable. The total radiation dose may affect RT planning, where it is important to comply with the spinal cord tolerance limit. Moreover, the integral body dose could vary. A possible explanation for the discrepancies between the results of our study and the previous series is the use of induction chemotherapy.

The limitations of this study included its retrospective, single-center design, which could have caused selection bias. Additionally, the number of patients was small and probably insufficient to reveal significant differences between the two groups in terms of short-term clinical endpoints including postoperative complications and survival outcomes. However, this study was meaningful in that nCRT with PBT for mainly upper to middle thoracic ESCC has rarely been reported. Late cardiac death occurred significantly more frequently in our patients with RT than in those who did not receive RT, as reported in a Surveillance, Epidemiology, and End Results analysis [24]. The effect of reducing cardiopulmonary toxicity in patients with EC and long-term survivors has not been studied. As the relationship between cardiac radiation dose and survival rate in patients undergoing definitive CRT in thoracic malignancies is still a hot topic [25], further research to reduce long-term cardiopulmonary toxicity in ESCC and EAC patients using PBT would be worthwhile. In this regard, the progress of the NRG GI-006 study is notable; we are also conducting a phase 2 study using proton therapy in nCRT for operable ESCC at our institution.

## 5. Conclusions

In conclusion, PBT provided a significant dosimetric benefit over photon-based RT by reducing radiation exposure to the lungs and heart in nCRT for operable ESCC. The dosimetric benefit must be validated in further clinical studies.

## Figures and Tables

**Figure 1 cancers-14-02033-f001:**
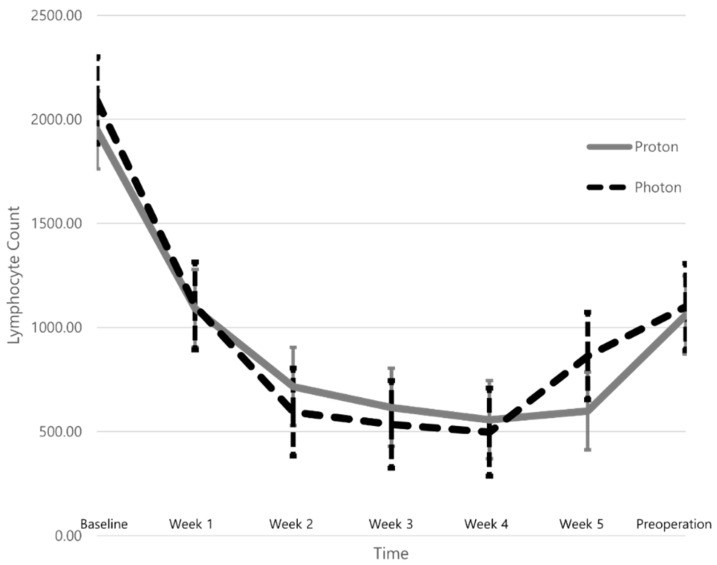
The trend of mean lymphocyte counts of patients with locally advanced esophageal squamous cell carcinoma, either Group_photon_ or Group_proton_, from the baseline through the neoadjuvant chemoradiotherapy, and until before the planned esophagectomy.

**Figure 2 cancers-14-02033-f002:**
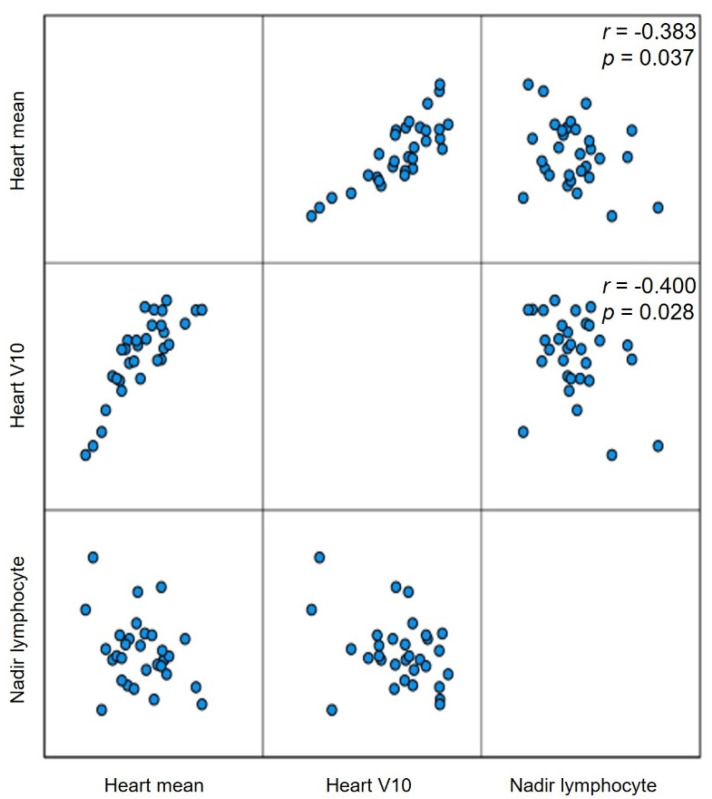
The correlation of the nadir of absolute lymphocyte count (Nadir lymphocyte) during the neoadjuvant chemoradiotherapy for locally advanced esophageal squamous cell carcinoma and the dosimetric parameters such as mean radiation dose to heart (Heart mean) and percentage of heart volume exposed to 10 Gy or more (Heart V10).

**Figure 3 cancers-14-02033-f003:**
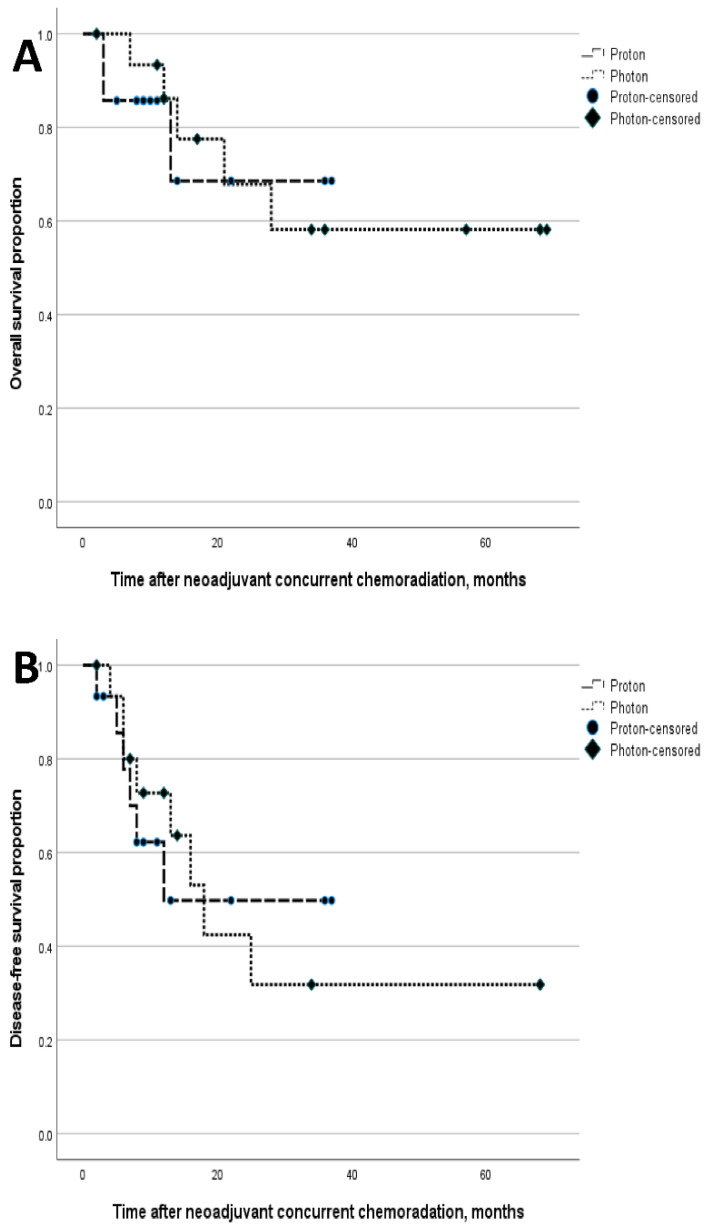
The survival results, overall survival (**A**) and disease-free survival (**B**) of locally advanced esophageal squamous cell carcinoma patients with neoadjuvant chemoradiotherapy according to the radiation modalities, photon, and proton.

**Table 1 cancers-14-02033-t001:** Patient characteristics of Group_photon_ and Group_proton_.

Variables	Group_photon_ (*n* = 16)	Group_proton_ (*n* = 15)	*p*-Value
Age (mean ± SD)	59.38 ± 4.92	62.27 ± 7.26	0.202
Sex (male, %)	14 (87.5)	13 (86.7)	1.000
Maximum SUV (mean ± SD)	12.71 ± 4.54 ^1^	13.18 ± 5.23 ^1^	0.800
ECOG PS (*n*, %)			0.043
0	0 (0.0)	4 (26.7)	
1	16 (100.0)	11 (73.3)	
Comorbidities			
None	10 (62.5)	5 (33.3)	0.104
Other cancer	1 (6.3)	3 (20.0)	0.333
Cardiovascular disease			
Hypertension	3 (18.8)	7 (46.7)	0.135
Arrhythmia	1 (6.3)	0 (0.0)	1.000
Coronary artery disease	0 (0.0)	1 (6.7)	0.484
Cerebrovascular disease	1 (6.3)	1 (6.7)	1.000
Asthma	0 (0.0)	1 (6.7)	0.484
Alcoholic liver cirrhosis	1 (6.3)	0 (0.0)	1.000
Diabetes	4 (25.0)	0 (0.0)	0.101
cT category (*n*, %) ^2^			0.326
1	1 (6.3)	1 (6.7)	
2	7 (43.8)	3 (20.0)	
3	7 (43.8)	11 (73.3)	
4	1 (6.3)	0 (0.0)	
cN category (*n*, %) ^2^			0.978
1	7 (43.8)	6 (40.0)	
2	8 (50.0)	8 (53.3)	
3	1 (6.3)	1 (6.7)	
Location (*n*, %)			0.412
Upper thoracic	5 (31.3)	4 (26.7)	
Middle thoracic	5 (31.3)	8 (53.3)	
Lower thoracic	6 (37.5)	3 (20.0)	

Abbreviations: SD = standard deviation, SUV = standard uptake value, ECOG PS = Eastern Cooperative Oncology Group performance status. ^1^ One missing value due to outside PET examination unable to interpret SUV. ^2^ American Joint Committee on Cancer staging manual 8th edition.

**Table 2 cancers-14-02033-t002:** Summary of dose volume histogram parameters and changes of lymphocytes during neoadjuvant chemoradiotherapy between Group_photon_ and Group_proton_.

Variables	Group_photon_ (*n* = 16)	Group_proton_ (*n* = 15)	*p*-Value
Total dose, Gy (median, range)	41.4 Gy (26.0–50.4)	41.4 Gy (37.8–50.0)	0.705
Lung mean, cGy (mean ± SD)	816.00 ± 251.63	431.27 ± 189.35	<0.001
Lung V10, % (mean ± SD)	23.65 ± 10.23	14.23 ± 6.21	0.005
Lung V20, % (mean ± SD)	13.31 ± 5.53	9.07 ± 3.87	0.02
Heart mean, cGy (mean ± SD)	2774.88 ± 651.92	1410.80 ± 615.92	<0.001
Heart V10, % (mean ± SD)	80.46 ± 13.68	55.11 ± 24.54	0.001
Heart V30, % (mean ± SD)	51.89 ± 21.31	18.35 ± 8.89	<0.001
Heart V40, % (mean ± SD)	35.43 ± 25.85	12.55 ± 6.40	0.003
Grade 4 lymphopenia (*n*, %)	2 (12.5%)	3 (20.0)	0.654
Lymphocyte count nadir (mean ± SD)	396.49 ± 156.48	388.02 ± 239.22	0.907

**Table 3 cancers-14-02033-t003:** Summary of surgical results for esophageal squamous cell carcinoma after neoadjuvant chemoradiotherapy according to the radiation modalities, photon, and proton.

Variables	Group_photon_ (*n* = 14) ^1^	Group_proton_ (*n* = 14) ^2^	*p*-Value
Minimal invasive surgery (*n*, %)	6 (42.9)	4 (28.6)	0.430
Intrathoracic anastomosis (*n*, %)	11 (78.6)	9 (64.3)	0.339
Stomach conduit (*n*, %)	14 (100.0)	13 (92.9)	1.000
LND			
3-field LND (*n*, %) ^3^	6 (42.9)	6 (42.9)	1.000
Resected LNs (median, range)	0 (0–44)	34 (0–66)	0.001
R0 resection	13 (92.9)	13 (92.9)	1.000
Tumor size, cm (median, range)	0.2 (0.0–13.0)	0.5 (0.0–3.1)	0.867
Pathologic stage (*n*, %) ^4^			
ypT0	5 (35.7)	4 (28.6)	0.544
ypTis	0 (0.0)	1 (7.1)	
ypT1	2 (14.3)	2 (14.3)	
ypT2	3 (21.4)	4 (28.6)	
ypT3	4 (28.6)	3 (21.4)	
ypN0	8 (57.1)	7 (50.0)	0.497
ypN1	5 (35.7)	7 (50.0)	
ypN2	1 (7.1)	0 (0.0)	
ypT0N0	5 (35.7)	4 (28.6)	1.000
ypM1	1 (7.1)	0 (0.0)	1.000
Operative complications (*n*, %)			
Anastomosis or graft failure			0.704
None	10 (71.4)	9 (64.3)	
Delayed anastomosis heal ^5^	1 (7.1)	2 (14.3)	
Endoscopic or surgical intervention ^6^	3 (21.4)	1 (7.1)	
Life-threatening mediastinitis ^7^	0 (0.0)	1 (7.1)	
Graft necrosis ^8^	0 (0.0)	1 (7.1)	
Respiratory complication			0.515
None	7 (50.0)	9 (64.3)	
Atelectasis requiring toileting BFS ^6^	6 (42.9)	5 (35.7%)	
ARDS ^7^	1 (7.1)	0 (0.0)	
Postoperative arrhythmia	1 (7.1)	3 (21.4)	0.596
Vocal cord palsy	5 (35.7%)	4 (28.6)	1.000
Chyle leak	0 (0.0)	2 (14.3)	0.481
Wound dehiscence	2 (14.3)	1 (7.1)	1.000
Operative mortality (*n*, %)	0 (0.0)	1 (7.1) ^9^	1.000

Abbreviations: ARDS = acute respiratory distress syndrome; LN = lymph node; LND = lymph node dissection. ^1^ One patient in Group_photon_ did not undergo curative resection after neoadjuvant chemoradiotherapy (nCRT) due to lack of resectability. ^2^ One patient in Group_proton_ did not undergo curative resection after nCRT due to lack of resectability, and the other due to progression of esophageal cancer. ^3^ All patients underwent either 2-field LND or 3-field LND. ^4^ American Joint Committee on Cancer staging manual 8th edition. ^5^ Clavien-Dindo classification grade I. ^6^ Clavien-Dindo classification grade III. ^7^ Clavien-Dindo classification grade IV. ^8^ Clavien-Dindo classification grade V. ^9^ One patient died on postoperative day 22 due to colon graft necrosis.

## Data Availability

Deidentified individual participant data are available and will be provided on reasonable request to the corresponding author.

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
