# Peer review of "A Comparative Analysis of Photon versus Proton Beam Therapy in Neoadjuvant Concurrent Chemoradiotherapy for Intrathoracic Squamous Cell Carcinoma of the Esophagus at a Single Institute"

_cancers, 2022, doi:10.3390/cancers14082033_

Round 1

Reviewer 1 Report

We suggest to define better the proton treatment process: beam-arrangement, positioning, use of planning ABC/4D CT, modality of treatment verification (CB-TC, use of fiducials marker), use or not of weekly 4D-CT scans

How do you score the postoperative complications? (Clavien-Dindo, CTCAE)

Author Response

Thank you for your valuable review.

My answers to your review are as follows. I hope you will give a good consideration after reading the reply.

Point 1: We suggest to define better the proton treatment process: beam-arrangement, positioning, use of planning ABC/4D CT, modality of treatment verification (CB-TC, use of fiducials marker), use or not of weekly 4D-CT scans

Response 1: We revised "Radiotherapy" part in Materials and Methods section by adding more details.

Point 2: How do you score the postoperative complications? (Clavien-Dindo, CTCAE)

Response 2: The summary of postoperative complications in Table 3 was intended to be based on classification by Clavien-Dindo. We revised Table 3 by adding notes to clarify its grade for "Anastomosis or graft failure" and "Respiratory complication" those which could vary in grades. As for other complications we omitted notes for better visualization: postoperative arrhythmia were all grade II, vocal cord palsy grade III, chyle leak grade II, wound dehiscence grade III. 

Reviewer 2 Report

This retrospective review compares proton vs photon rads as treatment in neoadjuvant setting for intrathoracic cancer. Though small and retrospective it suggests that proton therapy may be non-inferior to PBT, while providing an dosimetric effect.

As survival/disease outcome is a key part of the results, the authors should explain why ECOG zero patients were included in the proton analysis.  Though the study is small, ECOG status has a large impact on patient ability to tolerate treatment.  I would suggest a sensitivity analysis to see if ECOG status impacted outcomes.

Author Response

Thank you for your valuable review.

My answers to your review are as follows. I hope you will give a good consideration after reading the reply.

Point 1: As survival/disease outcome is a key part of the results, the authors should explain why ECOG zero patients were included in the proton analysis.  Though the study is small, ECOG status has a large impact on patient ability to tolerate treatment.  I would suggest a sensitivity analysis to see if ECOG status impacted outcomes.

Response 1: In our center, we consider ECOG 0 and 1 both equally as eligible candidates for esophagectomy after neoadjuvant concurrent chemoradiotherapy. Upon your opinion, we reviewed our results according to the ECOG status but did not find any significant impact on postoperative results and survival results.

Reviewer 3 Report

This manuscript describes neoadjuvant concurrent chemoradiotherapy (nCRT) for thoracic squamous cell carcinoma of the esophagus using either photon or proton beam radiation.

It is obvious that radiation dose to the lungs and heart can reduce by using proton beam compared with photon beam, and this might be beneficial for reducing late adverse events.

It is not surprising that 2-year overall survival and 2-year disease free survival were identical for both groups, because this study setting was nCRT. Surgical intervention was done for both groups. Srugery influenced the survival rates.

This reviewer thinks that to examine the difference of survival or disease free survival, the authors should compare definitive CRT setting (the authors write briefly in the discussion, but it should be emphasized).

Recent two randomized studies failed to show high radiation dose would result better outcomes. Therefore I would like to recommend to the authors that definitive CRT with photon dose of 50.4 Gy versus high dose proton dose around 70 Gy will be tested. Proton group at our country are now doing this type of study/survey.

Author Response

Thank you for your valuable review.

My answers to your review are as follows. I hope you will give a good consideration after reading the reply.

Point 1: This reviewer thinks that to examine the difference of survival or disease free survival, the authors should compare definitive CRT setting (the authors write briefly in the discussion, but it should be emphasized).

Response 1: This study population only included esophageal patients for neoadjuvant therapy. Comparing proton and photon for definitive CRT of esophageal cancer would be an interesting topic, which we are considering as another topic in near future. 

Point 2: Recent two randomized studies failed to show high radiation dose would result better outcomes. Therefore I would like to recommend to the authors that definitive CRT with photon dose of 50.4 Gy versus high dose proton dose around 70 Gy will be tested. Proton group at our country are now doing this type of study/survey.

Response 2: We appreciate you for the suggestion. We will give our thoughts on it for the future study.

Round 2

Reviewer 2 Report

Thank you for the response.  Good for publication.